# Intranasal Immunization with the Influenza A Virus Encoding Truncated NS1 Protein Protects Mice from Heterologous Challenge by Restraining the Inflammatory Response in the Lungs

**DOI:** 10.3390/microorganisms9040690

**Published:** 2021-03-26

**Authors:** Kirill Vasilyev, Anna-Polina Shurygina, Maria Sergeeva, Marina Stukova, Andrej Egorov

**Affiliations:** Smorodintsev Research Institute of Influenza, 197022 St. Petersburg, Russia; ann-polin@yandex.ru (A.-P.S.); mari.v.sergeeva@gmail.com (M.S.); stukova@influenza.spb.ru (M.S.); aevirol@gmail.com (A.E.)

**Keywords:** influenza virus, NS1 protein, immune response, immunogenicity, flow cytometry

## Abstract

Influenza viruses with an impaired NS1 protein are unable to antagonize the innate immune system and, therefore, are highly immunogenic because of the self-adjuvating effect. Hence, NS1-mutated viruses are considered promising candidates for the development of live-attenuated influenza vaccines and viral vectors for intranasal administration. We investigated whether the immunogenic advantage of the virus expressing only the N-terminal half of the NS1 protein (124 a.a.) can be translated into the induction of protective immunity against a heterologous influenza virus in mice. We found that immunization with either the wild-type A/PR/8/34 (H1N1) influenza strain (A/PR8/NSfull) or its NS1-shortened counterpart (A/PR8/NS124) did not prevent the viral replication in the lungs after the challenge with the A/Aichi/2/68 (H3N2) virus. However, mice immunized with the NS1-shortened virus were better protected from lethality after the challenge with the heterologous virus. Besides showing the enhanced influenza-specific CD8^+^ T-cellular response in the lungs, immunization with the A/PR8/NS124 virus resulted in reduced concentrations of proinflammatory cytokines and the lower extent of leukocyte infiltration in the lungs after the challenge compared to A/PR8/NSfull or the control group. The data show that intranasal immunization with the NS1-truncated virus may better induce not only effector T-cells but also certain immunoregulatory mechanisms, reducing the severity of the innate immune response after the heterologous challenge.

## 1. Introduction

Modifications of the influenza virus NS1 protein are considered a promising approach for the development of live-attenuated influenza vaccines and recombinant viral vectors. The effectiveness of such vaccine candidates in conferring protection against homologous and heterologous influenza strains was confirmed in different experimental models of influenza infection [1,2,3,4,5,6,7,8]. However, the immunological mechanisms providing heterologous protection are poorly understood.

The truncation of the NS1 protein is shown to increase the ability of the influenza virus to stimulate the innate immune cells and to enhance their antigen-presenting functions [9]. This leads to the formation of polyfunctional T-lymphocytes, which can elicit heterologous protection through the recognition of the broad spectrum of influenza antigens, including conserved epitopes [10,11].

An excessive immune response could be harmful to the host. Therefore, the regulatory mechanisms of the immune system must contribute to the protection of the organism after the secondary viral infection. The vaccination may not provide sterilizing immunity against antigenically mismatching pathogens, but the attenuation of the inflammatory immune response could protect the organism from tissue damage and ameliorate the course of the disease [12].

Here we investigated whether the immunogenic advantage of the virus expressing only the N-terminal part of the NS1 protein (124 a.a.) can be translated into the induction of the protective immunity against heterologous influenza virus in mice. To address this question, we compared the cross-protective efficacy of the immune response induced by intranasal immunization with the NS1-truncated influenza virus (A/PR8/NS124 (H1N1)) or by nonlethal infection with the wild-type (A/PR8/NSfull (H1N1)) influenza strain. To estimate the influence of immunization on the acute inflammatory immune response after the reinfection, we analyzed the concentrations of proinflammatory cytokines and the percentage of T-regulatory cells in mouse lungs after heterologous infection with the A/Aichi/2/68 (H3N2) strain.

## 2. Materials and Methods

### 2.1. Viruses

Two strains based on the influenza A/Puerto Rico/8/1934 (H1N1) virus were used for mice immunization: (1) A/PR8/NSfull virus, encoding the full-length NS1 protein; (2) A/PR8/NS124 strain, encoding the NS1 protein shortened to 124 a.a. The strains were created by “reverse genetics” [13], accumulated in developing chicken embryos (RCE) and purified by fractionation in a sucrose density gradient. The A/Aichi/2/68 (H3N2) influenza strain was used for the heterologous challenge.

### 2.2. Laboratory Animals

The study was performed on C57/Black-6 female mice obtained from the Biomedical Science Center (Stolbovaya, Russia). The experiments were conducted according to the guidelines for care and work with laboratory animals [14]. All of the animal studies were approved by the Russian Institutional Local Ethics Committee.

### 2.3. Immunization and Challenge Infection

Mice were immunized intranasally with 2.5 or 6 log_10_[TCID_50_] of A/PR8/NSfull and A/PR8/NS124 strains, respectively (30 µL of a virus suspension in sodium phosphate buffer (PBS, Biolot, St. Petersburg, Russia)) under light ether anesthesia. The control group received an equivalent volume of PBS. For the challenge infection, mice were inoculated with 2.5 LD_50_ of A/Aichi/2/68 (H3N2) at a volume of 30 µL. Bodyweight dynamics and the mortality rate were measured for 14 days after the challenge.

### 2.4. Virus Infectious Activity Analysis

To estimate the viral loads in mouse lungs, animals were sacrificed 2 and 4 days after the challenge with A/Aichi/2/68, and the lungs were homogenized using TissueLyser II (Qiagen, Hilden, Germany) bead homogenizer. Virus reproductive activity was determined by the titration of lung homogenates on MDCK cell culture. The assay was performed in 96-well culture plates (Nunc, Roskilde, Denmark) by the addition of 100 µL of prepared dilutions of the virus-containing material into the wells and subsequent incubation for 5 days at 34 °C and 5% CO_2_. The results were evaluated visually by cytopathic effect estimation and by a hemagglutination reaction (using 0.5% suspension of chicken red blood cells). Then, a 50% tissue culture infectious dose (TCID_50_) was calculated using the Reed and Munch method [15]. The viral titer was expressed as log_10_[TCID_50_]/mL.

### 2.5. Proinflammatory Cytokines Analysis

Concentrations of IFNα, IFNβ, IFNγ, IL-6, IL-1α, IL-1β, IL-27, IL-10, IL-23, TNFα, MCP-1, and GM-CSF in pg/mL were measured in lung tissue homogenates using LEGENDplex bead-based multiplex assay (Biolegend, San Diego, CA, USA) according to the manufacturer’s instructions. Samples were analyzed on a CytoFlex flow cytometer (Beckman Coulter, Brea, CA, USA), and the cytokine concentrations were calculated using LEGENDplex software, v. 8.0 (Biolegend).

### 2.6. Lymphocyte Isolation and Stimulation

T-cellular immune response was evaluated in the splenocytes and lungs of immunized mice 32 days postimmunization (d.p.im.) and in lungs 4 days postchallenge (36 d.p.im.). Mice were sacrificed by cervical dislocation. The chest cavity was opened, and the right ventricle was perfused with 10 mL of ice-cold DPBS (Biolot) before the removal of the lungs and spleen. Following the mechanical dissociation of the spleen with pestle homogenizers (Eppendorf, Hamburg, Germany) and collagenase/DNAse (Sigma, Saint Louis, Missouri, USA) digestion of lung tissue, cells were filtered through a 70 µm cell strainer. Erythrocytes were lysed with RBC lysis buffer (Biolegend) according to the manufacturer’s protocol. Cells were stained with antibodies for the innate immune populations’ analysis or seeded at a density of 1 × 10^6^ cells per well into flat-bottom 96-well tissue culture plates (Nunc) in RPMI 1640 (Gibco, Waltham, MA, USA) medium containing 10% fetal bovine serum (FBS, Gibco) and 1% penicillin/streptomycin solution (Gibco). To stimulate cytokine production, cells were incubated with 5 µg/mL of NP_366-374_ peptide and brefeldin A (Biolegend) for 6 h at 37 °C and 5% CO_2_. The peptide was synthesized by Verta Ltd. (Saint-Petersburg, Russia). Then, 10 mg of dry peptide were dissolved in 1 mL of DPBS (Biolot) and stored in small aliquots at −20 °C. The purity of the peptide was >90%, as determined by high-performance liquid chromatography.

### 2.7. Flow Cytometry

To analyze the innate immune cell populations, the panel of fluorochrome-conjugated antibodies was used, including CD11b-PE/Cy7, CD11c-PE, MHCII-Alexa488, CD103-PerCP-Cy5.5, CD45-APC/Cy7, CD64-BV421, and CD24-BV510. To estimate the percentage of CD8^+^ T-lymphocyte-producing cytokines after the peptide stimulation, cells were stained with CD8-PE/Cy7, CD4-PerCP-Cy5.5, CD44-BV510, CD62L-APC/Cy7, IFNγ-FITC, TNFα-BV421, and IL2-PE antibodies using the Fixation and Permeabilization Solution reagent kit (BD Biosciences) according to the manufacturer’s instructions. Zombie Red viability marker (BioLegend) was used to identify the dead cells. True Stain reagent, containing antibodies to CD16/CD32, was used to block nonspecific antibody binding (BioLegend). Data were collected on a CytoFlex flow cytometer (Beckman Coulter). The results were analyzed using the Kaluza Analysis v2 program (Beckman Coulter). To estimate the increase in the cytokine production levels upon the peptide stimulation, the background values obtained from the nonstimulated cells were subtracted from the corresponding values of stimulated samples before the statistical analysis.

### 2.8. ELISA

The levels of influenza-specific IgG antibodies were determined by ELISA using 96-well plates (NuncMaxisorp, Thermo Fisher Scientific, Waltham, Massachusetts, USA) covered by the A/Puerto Rico/8/1934 influenza virus accumulated in MDCK cell culture. The virus suspension at a concentration of 4 µg/mL in PBS (Biolot) was added to the wells and the plates were incubated for 12 h at 4 °C. Further, the plates were washed with 0.1% Tween20 solution (Thermo Fisher Scientific) and incubated for 2 h in PBS containing 5% FBS (Gibco) and 0.1% Tween20. The same buffer was used to prepare serial double dilutions of the analyzed sera samples. Plates were incubated for 1 h at room temperature. After extensive washing with 0.1% Tween20, secondary HRP-conjugated antibodies were added to the plates. A 100 µL volume of TMB substrate (KPL) was added to the wells, and the plates were incubated for 15 min. The development of a color reaction was stopped by the addition of 100 µL/well of 1N H_2_SO_4_. Optical density at a wavelength of 450 nm was measured using Synergy H1 Hybrid Multi-Mode Reader and analyzed with Gen5 software.

### 2.9. Statistical Analysis

RStudio Desktop 1.0.153 (RStudio Inc, Northern Ave, Boston, USA) was used for statistical data analysis. The Dunnett test was used to compare several experimental groups with one control group. To compare two experimental groups, the Student’s *t*-test was used. Multiple comparisons were performed using univariate analysis of variance (ANOVA) followed by pairwise comparison of groups using the Tukey criterion.

## 3. Results

### 3.1. Humoral and T-Cellular Immune Response to Immunization with A/PR8/NS124 or A/PR8/NSfull Influenza Virus

It is shown [9] that A/PR8/NS124 and A/PR8/NSfull strains differ significantly in their pathogenicity and reproduction activity in mouse lungs. For immunization with the A/PR8/NSfull virus, we selected the dose of 2.5 log_10_[TCID_50_] as the highest nonlethal dose. Mice immunized with the A/PR8/NS124 strain received a sublethal dose of the virus equal to 6 log_10_[TCID_50_]. To analyze the humoral immune response, blood samples were collected 21 days postimmunization (d.p.im.), and influenza-specific IgG antibody levels were estimated with ELISA (Figure 1) and confirmed in the hemagglutination inhibition (HAI) test (data not shown). Average HAI titers were 1:128 in both NS124 and NSfull groups. The T-cell immune response was measured in lungs and spleen 32 d.p.im.

The analysis of humoral immune response revealed that the inoculation of mice with 2.5 log_10_[TCID_50_] of A/PR8/NSfull and 6 log_10_[TCID_50_] of A/PR8/NS124 induced comparable levels of influenza-specific IgG antibodies. To estimate the level of the T-cellular immune response, we calculated the percentage of CD8^+^ effector memory T-lymphocytes (Tem) producing any combination of IFNγ, IL2, and TNFα in response to in vitro stimulation with the NP_366-374_ peptide. A pronounced systemic (spleen) and local (lungs) CD8^+^ T-cellular immune response was formed in both groups at 32 d.p.im. Most of the antigen-specific T-cells were represented by double-positive (IFNγ^+^IL2^−^TNFα^+^) and triple-positive (IFNγ^+^IL2^+^TNFα^+^) cytokine-producing CD8^+^ T-lymphocytes both in the lungs and spleen. The A/PR8/NS124 virus induced the formation of higher numbers of polyfunctional IFNγ^+^IL2^+^TNFα^+^ Tem (*p* = 0.02), but no significant difference between the experimental groups was shown in the total content of cytokine-producing cells as well as in the level of single-positive or double-positive T-lymphocytes. Thus, based on the results of humoral and T-cellular immune response analysis, we conclude that the proposed immunization scheme allowed the induction of similar levels of influenza-specific immunity in both NSfull and NS124 groups.

### 3.2. Immunity Triggered by A/PR8/NS124 Prevents Mortality after the Heterologous Challenge

To determine the cross-protection efficiency of the acquired immune response, mice were challenged with 2.5 LD_50_ of the A/Aichi/2/68 (H3N2) virus 32 d.p.im. The mortality and the bodyweight dynamics were monitored for 14 days after the heterologous challenge (d.p.c.). Proinflammatory cytokine concentrations and viral loads were measured in lungs homogenates 2 and 4 d.p.c. The percentage of T-regulatory cells and the level of influenza-specific CD8^+^ T-cellular immune response were estimated in lungs 4 d.p.c. The relative content of innate immune cell populations was analyzed right before heterologous infection (0 d.p.c.) and on 1 and 4 d.p.c.

The results of mortality, body weight dynamics, and viral load analyses are represented in Figure 2. Mice infected with 2.5 log_10_[TCID_50_] of the A/PR8/NSfull virus, as well as nonimmunized animals, were characterized by a bodyweight reduction up to 80–75% of the initial values during 8 d.p.c. In both control and NSfull groups, the mortality rate was about 40%. At the same time, intranasal immunization with 6 log_10_[TCID_50_] of A/PR8/NS124 completely abolished mortality and significantly reduced weight loss in mice (Figure 2A,B).

It is noteworthy that immunization did not prevent infection with A/Aichi/2/68 and the viral replication. Figure 2C shows that viral titers in lung homogenates in both immunized and nonimmunized mice were about 6.16–6.79 log_10_[TCID_50_]/mL 2 d.p.c. There was no significant difference between the experimental groups. The viral load decreased in the NS124 group in comparison to the control (*p* = 0.03) 4 d.p.c., but the difference between NS124 and NSfull groups remained insignificant. Thus, immunization with A/PR8/NS124 reduced the severity of the disease and contributed to the survival of infected animals by the mechanisms that do not rely on the control over the viral infection.

### 3.3. Intranasal Immunization with the A/PR8/NS124 Virus Restrains the Inflammatory Response in the Lungs after the Heterologous Challenge

The better protection after immunization with the A/PR8/NS124 virus could be explained by the attenuation of the inflammatory response in the lungs of challenged animals. To test this hypothesis, we measured the concentration of inflammatory cytokines in lung homogenates two and four days after the infection with the A/Aichi/2/68 virus. We also estimated the dynamics of the innate immune cell populations on 0, 1, and 4 d.p.c. The results are shown in Figure 3.

The analysis of proinflammatory cytokine concentrations in the lung homogenates of mice showed that immunization with A/PR8/NS124 led to a substantial reduction in the cytokine response compared to the control group 2 d.p.c. On the contrary, the infection with A/PR8/NSfull was not accompanied by a decreased inflammatory response after the heterologous challenge. The greatest difference between the NSfull and NS124 groups was associated with the levels of type I IFNs (Figure 3A). The concentrations of IFNα and IFNβ were on average 22 and 10 times (respectively) lower in the NS124 group than in the NSfull group (*p* = 0.04; *p* = 0.001). In addition, the significant difference between A/PR8/NS124- and A/PR8/NSfull-immunized mice 2 d.p.c. was shown in the levels of IL-6 (*p* = 0.05), IL-27 (*p* = 0.02), IL-23 (*p* = 0.01), TNFα (*p* = 0.05) and GM-CSF (*p* = 0.002).

At 4 d.p.c., a slight increase in the concentrations of the cytokines was observed in the NS124 group, and the difference from the NSfull group became less prominent. Both NSfull and NS124 groups showed a significantly increased IFNγ concentration compared to the control (*p* = 0.01, 0.05, respectively). Similarly, an increased IL-10 level was observed in both NSfull and NS124 groups, which may indicate more active T-regulatory lymphocytes in the immunized animals (Figure 3A). The obtained data confirm the hypothesis that the survival of mice in the NS124 group after heterologous infection was higher due to a decreased inflammatory response. The reduced level of MCP-1 and GM-CSF in the NS124 group, in comparison to the control and the NSfull group 4 d.p.c., allows expecting a decrease in lung infiltration by the innate immune cells, such as neutrophils and macrophages.

To estimate the difference in the relative content of innate immune cell populations in the NSfull, NS124, or control groups before and after the heterologous challenge, processed lung tissue was stained with fluorescently labeled CD45, MHCII, CD11b, CD11c, CD24, CD64, and Ly6G antibodies. In comparison to the control group, the level of alveolar macrophages (AMP) retained in the lung tissue of A/PR8/NS124-immunized mice increased 32 days after immunization (0 d.p.c.). Moreover, 24 h after the challenge with the A/Aichi/2/68 virus, the NS124 group showed a statistically significant increase in the relative content of these cells compared to the control (*p* = 0.0006) and the NSfull group (*p* = 0.0045). However, the percentage of AMP was similar in all three groups 4 d.p.c. (Figure 3B).

The most prominent difference between the control, NSfull, and NS124 groups was associated with the neutrophils and interstitial macrophage (IMP) dynamics. Nonimmunized animals showed a three-fold increase in the relative content of these populations in the lungs. Mice in the NSfull and NS124 groups also showed a slight increase in the number of neutrophils and IMP, but the values did not differ significantly from Day 0 in these groups. The percentage of IMP in the control group was 26.91 ± 0.98% of total CD45^+^ cells 4 d.p.c., whereas only 15.73 ± 1.28% (*p* = 0.0002) and 14.30 ± 2.18% (*p* = 0.0003) of lung CD45^+^ cells were represented by IMP in the NSfull and NS124 groups, respectively. Interestingly, despite the higher concentrations of proinflammatory cytokines in the NSfull group compared to the NS124 group, no difference was shown between these groups in the IMP percentage. The percentage of neutrophils 4 d.p.c. was 19.49 ± 0.82% in the control group, while it was 10.56 ± 0.95% (*p* = 0.0002) and 7.29 ± 1.07% (*p* = 0.0002) in the NSfull and NS124 groups, respectively. It should be noted that the number of neutrophils was also significantly higher in the NSfull group compared to the NS124 group 4 d.p.c. (*p* = 0.03). The decreased neutrophilic infiltration of lungs in the NS124 group is likely to be associated with the attenuation of immunopathology after the heterologous influenza challenge.

### 3.4. Enhanced CD8^+^ T-Cellular Immune Response in A/PR8/NS124-Immunized Mice after the Heterologous Challenge

To estimate the antigen-specific T-cellular immune response in the immunized mice after heterologous infection with A/Aichi/2/68, cells from lung homogenates were stimulated with the NP_366-374_ peptide 4 d.p.c. (Figure 4). The percentage of cytokine-producing T-cells was evaluated with flow cytometry.

As shown in Figure 4, mice immunized with A/PR8/NS124 had a higher total percentage of cytokine-producing CD8^+^ effector T-lymphocytes compared to the NSfull group (*p* = 0.002). It should be emphasized that the difference between the NS124 and NSfull groups was much more pronounced four days after infection with A/Aichi/2/68 compared to the baseline (Figure 1A). Heterologous infection led to an increased CD8^+^ T-cellular immune response in the NS124 group but not in the NSfull group, where the proportion of cytokine-producing T-lymphocytes remained the same as before the challenge. In both groups, polyfunctional T-lymphocytes (IFNγ^+^IL2^−^TNFα^+^ and IFNγ^+^IL2^+^TNFα^+^) dominated the immune response, but the percentages of these populations were approximately three times higher in the NS124 group than in the NSfull group (*p* = 0.004; 0.00003).

Thus, both A/PR8/NSfull and A/PR8/NS124 viruses induced comparable levels of T-cell immune response in the lungs after intranasal immunization. The intensity of the response decreased over time in both groups. However, after the reinfection with the heterologous virus, A/PR8/NS124-immunized mice demonstrated the increased reactivity of antigen-specific CD8^+^ T-lymphocytes.

### 3.5. Induction of T-Regulatory Cells in Mouse Lungs after the Heterologous Challenge

Immunization with the NS1-truncated influenza virus attenuated the severity of the secondary innate immune inflammatory reaction but enhanced the T-cellular response to the immunodominant NP_366-374_ epitope after the heterologous challenge. The lower weight loss and the zero mortality rate, as well as the reduced concentrations of proinflammatory cytokines, may indicate the enhanced activity of the regulatory mechanisms of the immune system in the NS124 group compared to the NSfull or control groups. To address this hypothesis, we analyzed the relative content of T-regulatory cells in the lungs of experimental animals four days after the challenge. A significant increase in the proportion of FOXP3^+^Helios^+^ T-regs was shown in both NSfull and NS124 groups compared to the control (*p* = 0.007; 0.001, respectively, Figure 5). The data are consistent with the increased concentration of regulatory cytokine IL-10 observed in the lung homogenates of mice from the NS124 (*p* = 0.01) and NSfull (*p* = 0.008) groups 4 d.p.c. compared to the control (Figure 5). No significant difference was shown between the two experimental groups in the percentage of T-regulatory cells; however, in the NS124 group, the number of T-regs was higher than in the NSfull group.

## 4. Discussion

Existing inactivated and live-attenuated flu vaccines should be renewed every year due to extremely high evolutionary variability and genetic heterogeneity of circulating influenza viruses. Research mostly considers adaptive immunity to the conserved antigens of the influenza virus as a major factor of cross-protection. Inactivated influenza vaccines primarily induce a humoral immune response, providing effective protection only when the antigenic structure of surface influenza proteins in vaccine strains coincides with the one of circulating viruses [16]. Live-attenuated influenza vaccines (LAIVs) induce both humoral and T-cellular immunity due to their capacity to replicate in the upper respiratory tract and stimulate the MHC-I/II-dependent antigen presentation. Influenza viruses of A, B, and C types share some common CD4/8^+^ T-cellular epitopes, which could induce the cross-protective immune response [17]. However, the T-cellular immune response induced by LAIVs is insufficient to provide broader cross-protection [18].

Modern variations of universal vaccines, such as HA stalk-based constructions [19,20,21,22], peptide vaccines [23,24], DNA/RNA-vaccines [25,26,27,28,29,30], or artificial HA-proteins created using Computationally Optimized Broadly Reactive Antigen (COBRA) technology [31], are focused on the induction of antibodies and effector T-cells, recognizing the conserved epitopes of the influenza virus. However, the immune response to the restricted set of the conserved epitopes does not prevent the formation of the escape mutants. Even the conserved stalk regions of HA can acquire mutations that lead to the formation of virulent strains resistant to cross-reactive human antibodies [32,33]. The selective host immune system pressure reduces the number of epitopes per virus due to the accumulation of mutations primarily in the immunogenic regions of viral proteins [34]. It is shown that, since 1968, the total number of the T-cellular epitopes of circulating influenza strains of the H3N2 subtype has been decreasing with the rate of more than one epitope per three years. The overall HLA binding affinity of influenza epitopes has been decreasing as well [34]. It is, therefore, important to pay attention to the nonspecific protection mechanisms related to the phenomenon known as the “innate immune memory” or “trained innate immunity”, which is induced by influenza vaccines [35,36,37,38,39,40]. The activity of the immune system regulatory component should also be considered in the context of the postvaccine immune response to the heterologous viral strains. As shown by several studies, mortality, induced by the influenza virus, arises from the immunopathology and cytokine storm rather than directly from the viral burden [41]. The balance between the effector and regulatory components of the immune response and their coordination provides the elimination of the pathogen and prevents immunopathology.

In the present work, we compare the heterologous immune response induced either by immunization with the NS1-truncated influenza virus or by the nonlethal infection with the wild-type A/PR/8/34 influenza strain. Previous studies have shown that the shortening of the NS1 protein enhances the innate and adaptive (T-cellular and humoral) immune response to the influenza virus [1,9,11]. Cross-protection against heterologous influenza strains in mice immunized with the NS1-modified influenza viruses is also reported [4,6,42]. To investigate the mechanisms of heterologous protection, we immunized mice with the A/PR8/NS124 or A/PR8/NSfull virus and challenged them with the A/Aichi/2/68 influenza strain 32 days after immunization. The doses of viral strains, chosen for primary infection, induced comparable levels of humoral and T-cellular immune response.

The A/PR8/NS124-immunized mice demonstrated better protection against A/Aichi/2/68 compared to those preinfected with the A/PR8/NSfull virus. Immunization did not prevent infection with A/Aichi/2/68 in the NSfull or NS124 groups. No statistically significant difference was found in viral titers between these groups two and four days after the challenge. However, the levels of proinflammatory cytokines in the lungs, such as IFNα/β, IL-6, IL-23, GM-CSF, were lower in the NS124 group compared to the NSfull and control groups. Since the hyperproduction of IFN-I is the major cause of immunopathology [43,44], the decrease in the inflammation intensity could be assumed to have prevented the lethality in the NS124 group. We also showed that immunization with the NS1-truncated influenza virus prevented the excessive neutrophil infiltration of the lungs upon heterologous infection.

On the other hand, immunization with the A/PR8/NS124 virus led to a more robust T-cellular immune response to the NP_366-374_ epitope after the heterologous challenge compared to the NSfull group. Even though both NSfull and NS124 groups mount a similar T-cellular immune response to the in vitro stimulation with the NP_366-374_ 32 d.p.im., the A/PR8/NS124-immunized mice showed a three-fold increase in the number of antigen-specific T-cells after the heterologous challenge, while in the NSfull group, the level of antigen-specific T-lymphocytes did not change compared to 32 d.p.im. We assume that the truncation of the NS1 protein intensified the impact of the influenza virus on both the effector and the regulatory component of the immune response. The advantages of A/PR8/NS124-induced immunity could be realized through several mechanisms, including the modulation of the innate immune response, the formation of more effective T-regulatory cells, and the generation of local antigen-specific T-lymphocytes.

The analysis of innate immune cell populations in the NSfull and NS124 groups revealed an increase in the percentage of AMP in the lungs of mice, immunized with the A/PR8/NS124 virus, on the first day after A/Aichi/2/68 infection. The increase is accompanied by the attenuated inflammatory reaction 2 d.p.c. The alveolar macrophages are located in the airways and normally produce the regulatory cytokines, such as IL-10 and TGFβ [45]. It is known that depletion of the AMPs before viral infection leads to the burst of virus replication and a significant aggravation of the disease [46]. AMP deficiency is also shown to promote immunopathology in mice [47]. Thus, it is possible that the influenza virus with the truncated NS1 protein stimulated the influenza virus AMPs migration to the airways and induced the activated state of these cells. After the challenge infection, AMPs mounted an increased anti-inflammatory response, preventing immunopathology.

The analysis of the T-regulatory cell levels in the lungs of immunized mice after the heterologous challenge revealed an increase in the percentage of FOXP3^+^Helios^+^ T-regs in both experimental groups in comparison to the control. According to the literature data [48], FOXP3^+^Helios^+^ T-regs are characterized by the most pronounced immunosuppressive activity compared to other regulatory lymphocyte populations. Despite the absence of a statistically significant difference in the percentage of these cells between the NSfull and NS124 groups, the T-regulatory lymphocytes in the A/PR8/NS124-immunized mice are possibly more active and prevent excessive inflammation and lung infiltration through the mechanisms, which has not been studied here.

NS1-truncated influenza viruses are strong IFN-I-inducers [49,50,51]. However, an increase in the IFN-I concentration upon intranasal immunization with these viruses is transient due to their restricted ability to proliferate in the lung tissue. Short-term exposure to IFN-α/β is shown to enhance the subsequent T-cell proliferation [52,53]. On the other hand, prolonged exposure to IFN-α/β has an antiproliferative effect [54]. We assume that immunization with the A/PR8/NS124 virus induces higher proliferative potential in the influenza-specific T-lymphocytes that results in an increased response to the subsequent infection.

It could be concluded that the impairment of the NS1 protein activity promotes not only the enhanced immunogenicity of the T-cellular epitopes of the influenza virus but also better regulation of the inflammatory immune response to the heterologous challenge. All the modern live influenza vaccines approved for clinical usage express the full-length NS1 protein, which may jeopardize their advantages over the inactivated flu vaccines. We claim that the truncation of the NS1 protein can be used to create the highly immunogenic influenza vaccines inducing the cross-protection through the activation of both the effector and regulatory component of the immune system.

## Figures and Tables

**Figure 1 microorganisms-09-00690-f001:**
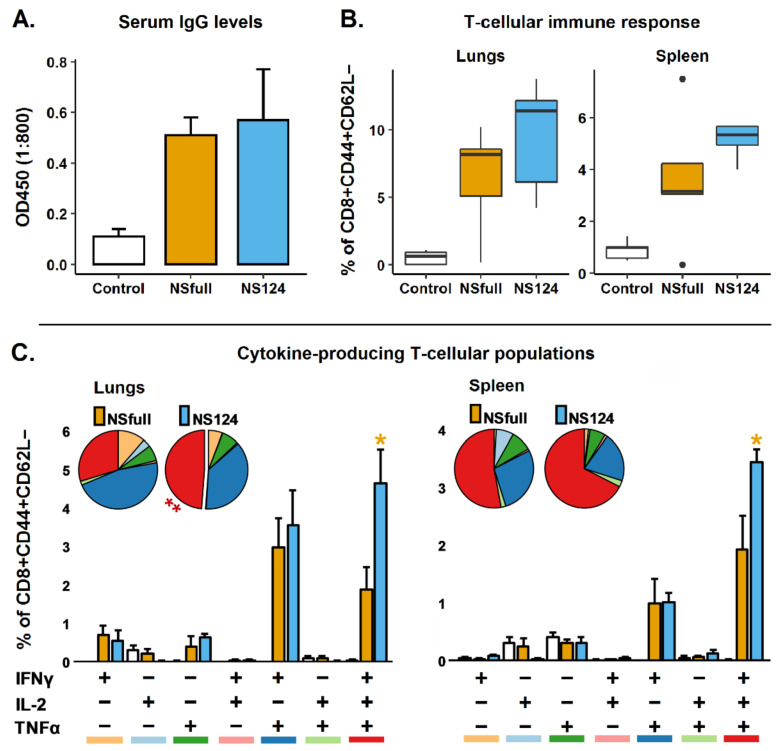
Humoral and CD8^+^ T-cellular immune response to intranasal immunization with A/PR8/NSfull or A/PR8/NS124 influenza strains. (**A**). Total level of influenza-specific IgG determined with ELISA. The histograms represent the optical density of the serum samples diluted 1:800 (mean ± SD, *n* = 8). (**B**). Total percentage of cytokine-producing effector memory CD8^+^CD44^+^CD62L^−^ T-lymphocytes after 6 h of in vitro lung or spleen cells stimulation with the NP_366-374_ peptide 32 d.p.im. (**C**). Histograms represent the percentage of different populations of cytokine-producing T-cells in the total effector memory (EM) CD8^+^CD44^+^CD62L^−^ T-lymphocytes population. Pie charts represent the percentage of cells producing any combination of IFNγ, IL2, or TNFα cytokines in the total cytokine-producing CD8^+^ EM T-cell subset. Color bars under histogram mark different cytokine-producing populations and correspond to the pie chart color scheme. Groups were compared using ANOVA followed by Tukey’s post-hoc test (*: *p* < 0.05, **: *p* < 0.001, *n* = 4).

**Figure 2 microorganisms-09-00690-f002:**
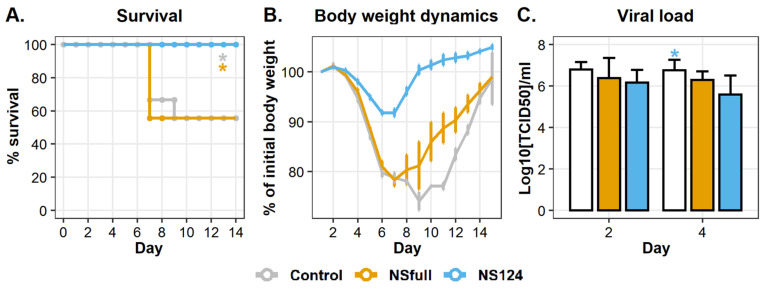
Survival dynamics, body weight dynamics, and lung viral titers of mice intranasally immunized with A/PR8/NSfull and A/PR8/NS124 viruses after the heterologous challenge with A/Aichi/2/68. (**A**). The Kaplan–Meier curves representing the percentage of survived animals at different time points after the challenge (*: *p* < 0.05, the curves were compared using the Cox method, *n* = 6). (**B**). Bodyweight dynamics. (**C**). Viral load (log_10_[TCID_50_]/mL, mean ± SD, *n* = 4; *: *p* < 0.05. Groups were compared using ANOVA followed by Tukey’s post-hoc test.

**Figure 3 microorganisms-09-00690-f003:**
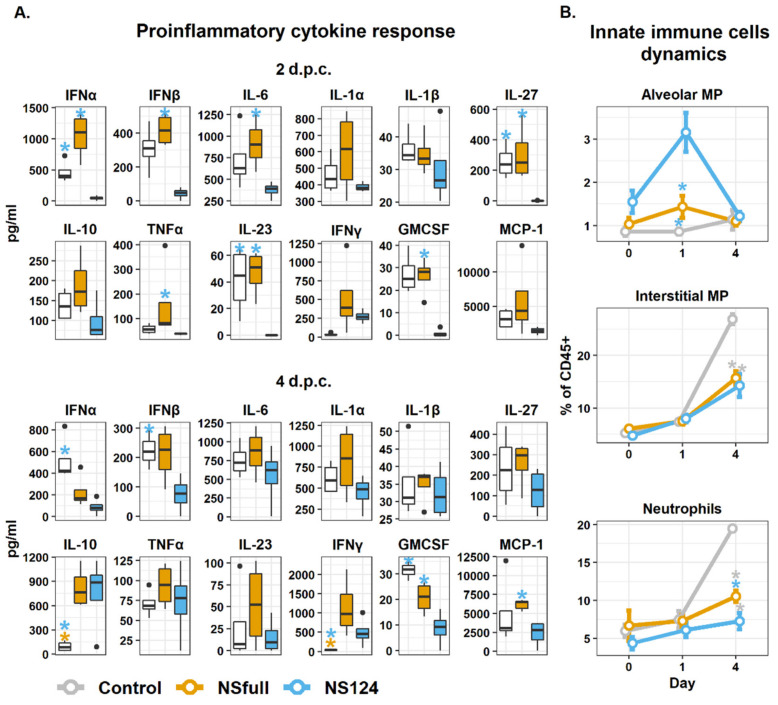
Proinflammatory cytokine response and innate immune cells’ dynamics in mice intranasally immunized with the A/PR8/NSfull and A/PR8/NS124 viruses after the heterologous challenge with A/Aichi/2/68. (**A**). Concentration of cytokines in lung tissue homogenates 2 and 4 days after the heterologous challenge (*: *p* < 0.05. Groups were compared using ANOVA followed by Tukey’s post-hoc test). (**B**). Percentage of alveolar macrophages, interstitial macrophages, and neutrophils in the lungs of immunized mice 0, 1, and 4 days after the heterologous challenge (mean ± SEM, *n* = 4, *: *p* < 0.05. Groups were compared using ANOVA followed by Tukey’s post-hoc test).

**Figure 4 microorganisms-09-00690-f004:**
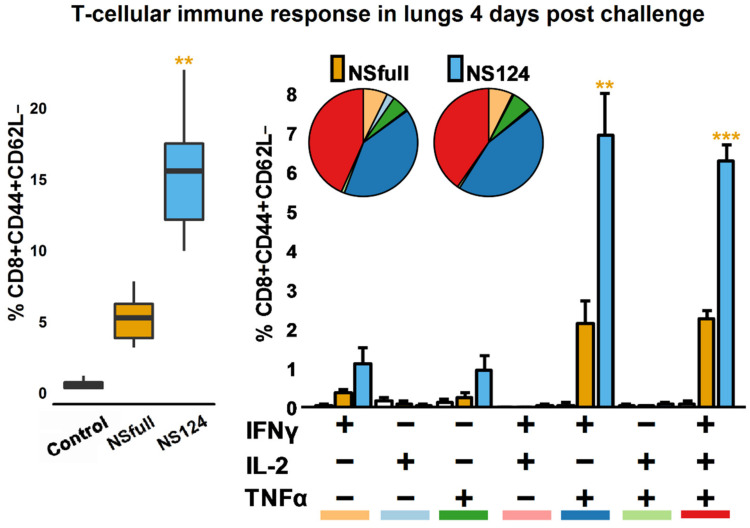
CD8^+^ T-cellular response in mice immunized with the A/PR8/NSfull or A/PR8/NS124 influenza strain after the challenge with A/Aichi/2/68. Box plots represent the total percentage of cytokine-producing effector memory CD8^+^CD44^+^CD62L^−^ T-lymphocytes after 6 h of in vitro lung or spleen cells stimulation with the NP_366-374_ peptide 4 d.p.c. Histograms represent the percentage of different populations of cytokine-producing T-cells in the total effector memory (EM) CD8^+^CD44^+^CD62L^−^ T-lymphocytes population. Pie charts represent the percentage of cells producing any combination of IFNγ, IL2, or TNFα cytokines in the total cytokine-producing CD8^+^ EM T-cell subset. Color bars under histogram mark different cytokine-producing populations and correspond to the pie chart color scheme. Groups were compared using ANOVA followed by Tukey’s post-hoc test (**: *p* < 0.001, ***: *p* < 0.0001, *n* = 4).

**Figure 5 microorganisms-09-00690-f005:**
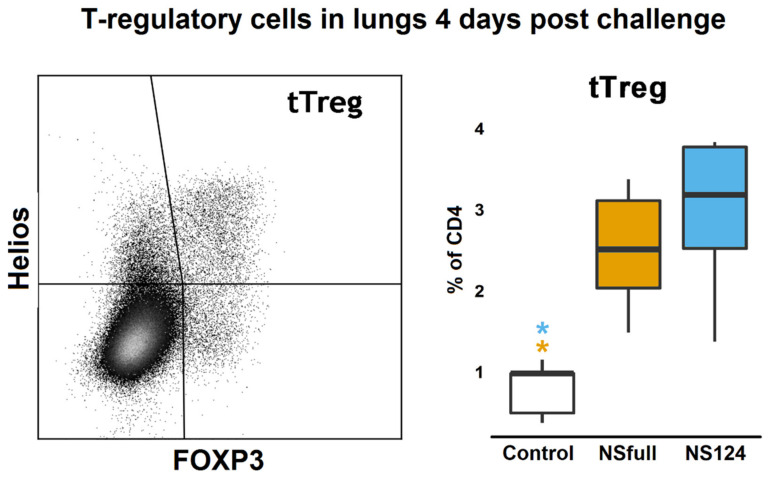
T-regulatory cells in the lungs of mice intranasally immunized with the A/PR8/NSfull and A/PR8/NS124 viruses after the heterologous challenge with the A/Aichi/2/68. The right plot represents the percentage of FOXP3^+^Helios^+^ T-regs in mouse lungs 4 d.p.c. *: *p* < 0.05. Groups were compared using ANOVA followed by Tukey’s post-hoc test.

## Data Availability

The data presented in this study are available on request from the corresponding authors.

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
