# Peer review of "Intranasal Immunization with the Influenza A Virus Encoding Truncated NS1 Protein Protects Mice from Heterologous Challenge by Restraining the Inflammatory Response in the Lungs"

_microorganisms, 2021, doi:10.3390/microorganisms9040690_

Round 1

Reviewer 1 Report

The work “Intranasal immunization with the influenza A virus encoding truncated NS1 protein protects mice from heterologous challenge by restraining the inflammatory response in the lungs” by Vasilyev et al is a thorough examination of the immune response following heterologous challenge to NS1 full and truncated NS1 vaccine strains. The authors present a compelling argument for the use of a shortened NS1 since it was correlated with reduced immunopathology compared with the NS1 full strain, while levels of protection were similar. I have only minor suggestions for revision.

Figure 1- the chart titles for A and B should be more descriptive and match what is written in the legend- Ie levels of IgG instead of Antibody immune response etc.

The header for section 3.4 contains a typo “CB8.”

Author Response

Thank you for your suggestions.

Additional clarifications are introduced in figures 1-5:

Figure 1: Title of figure 1A changed from "Antibody immune response" to "Serum IgG levels". The title "Cytokine-producing T-cellular populations" added to figure 1C.

Figure 2: Titles of axis x added to figures 2A-C.

Figure 3: Titles "Proinflammatory cytokine response" and "Innate immune cells dynamics" added to figures 3A, 3B.

Figure 4: Title "T-cellular immune response in lungs 4 days post challenge" is added.

Figure 5: Title "T-regulatory cells in lungs 4 days post challenge" is added.

The typo in the header for section 3.4 is fixed.

Reviewer 2 Report

I have now read the manuscript titeled “Intranasal immunization with the influenza A virus truncated NS1 protein protects mice from heterologous challenge by restraining the inflammatory response in the lung” by Vasilyev K et al. (microorganisms-1141305).

The manuscript present the systemic as well as the lung tissue immune responses with their influenza A viruses and mainly their obtained cell-mediated immune response results.

The study set up and design is interesting and provides several preliminarily interesting results and data. The limitations of the presented study with these interesting LAIV vaccine candidates seem to be the relatively small study group sizes and short-term immunity development that may explain that the some of the immune response results and heterologous protective capacity is limited.

Comments and questions:

The study is of significant interest when it comes to the presentation of the immunological responses and infectious virus levels between the study groups.

Q1. The limitations in the study are high-lighted by the authors in their introduction of their study as an intranasal immunization study. The immunization and challenge volumes are more than intranasal, it was probably also oral, lung and gastrointestinal? The results may be rather based on a general mucosal viral immunization flushing, since the used volume used will more than fill the nasal tissues and environment.

Q2. Materials and Methods:

The authors call their immunization design Intranasal immunization. Line 66-71.  However, the volume (30 ul vaccine) is very large (this volume would represent a volume of about seven liters of vaccine flushing in a human nose?) to only target the intranasal immune system. Rather this volume would probably reach all mucosal compartments from nasal, oral, low respiratory tract, lungs, and perhaps even gastrointestinal tract immune systems? The authors should explain their thoughts and arguments behind their mucosal immunization design more in depth.

Q3. Line 66-71. The influenza virus challenge volume used was 30 ul virus volume dose. Was this done to reach the lung tissue with a greater volume of virus?

Q4. It is not clear if the mice were sedated or not during immunizations and challenges. The authors should clarify this.

Results:

Q5a. In Figure 1 A the authors show the serum IgG reactivity to coated influenza A virus. It is not clear toward which influenza A viral proteins this reactivity is directed? It would be good if the authors could comment on this.

Q5b. Were the detected antibodies in serum capable of neutralizing influenza viruses?

Q5c. Is the OD450 (1:800) antibody level considered high or low? Serum samples with known anti-influenza A IgG reactivity would have been valuable to include as controls.

Q5d. How certain are the authors on the anti-influenza A viral IgG reactivity ?

 It would be good if the authors could present these serum reactivities to a control antigen (egg-suspension without virus).

Q6. The Figure 1 B and Figure 1C seem to be based on studies performed with four individuals (n=4). Line 160? In Figure 1A the numbers of tested animals seem to be 8 (n=8)?  Why were not all 8 animals included in the T-cell analyses of the lung and spleen analyses? Which 4 animals were excluded?

Q7a. Figure 2C. should the X-axis state days or numbers of tested mice?

Q7b. Figure 2. The differences in survival and viral loads between groups immunized with influenza A/PR8/NSfull and A/PR8/NS124 are statistically very small. It is possible that this may be due to the study group size of mice (n=4 per group?, thus 1 individual represents 25% of the study participants?). The authors should comment on this limitation, or preferably perform a repetition study with a few more animals/group.

Q8. Discussion:

Line 328-334. The authors question the general selection criteria for selecting seasonal influenza vaccines based on the adaptive immune responses.

Instead, the authors suggest that the innate immune parameters should be considered.

However, the selection of functional vaccines is more reproducible on group level when the old fashioned adaptive immune response pattern is used than the innate immune reaction patterns that may be considerably more individual and varying, thus making this side of the immune reactivity very difficult to use, except with selected individuals are evaluated, rather than on a groups level.

The study design, to consider the authors argumentation should have needed to be performed on a heterologous study population of mice, instead of on an inbred C57-animal strain?

Author Response

Thank you for your suggestions and comments.

Answers to Q1-Q2. The volume of 30 mkl was used as sufficient for the induction of the system immune response and local immune response in mouse lungs. The goal of the research was to investigate the local immune reactions in lung tissue of the vaccinated animals. Thus, the applied immunization strategy was intended to induce not only intranasal immune response but also to stimulate the immune response in the lungs. The term “intranasal immunization/vaccination” is commonly used in the literature to describe the process of immunization of mice with live attenuated influenza vaccines in a volume of 20-50 mkl.

Answers to Q3. Chosen conditions were optimal for the induction of severe lung infection in mice.

Answer to Q4. Light ether anesthesia was used to sedate mice during immunization and challenge. Changes are introduced in the “materials and methods” section (line 68).

Answer to Q5a. Whole viral (A/PR/8/34) particles were used as a target antigen for the ELISA. So there is no information about the specificity of the serum IgG to particular influenza proteins.

Answer to Q5b. The data obtained with ELISA were confirmed with the hemagglutination inhibition test (data not shown in the article). Average HAI titers were about 1:128 in both NS124 and NSfull groups and no statistically significant differences between groups were shown. According to the literature, the values obtained with the hemagglutination inhibition test and neutralization test are perfectly correlated, so we could expect the neutralizing activity of antibodies in serum from both groups. In this work, we analyzed the serum antibody level in immunized mice only to confirm that the chosen immunization strategy induces a similar magnitude of the immune response in both groups.

Answer to Q5c. The OD450 values obtained with ELISA correspond to HAI titers about 1:128 in both groups. Additional information on HAI test results is added to the article (lines 148-150).

Answer to Q5a. For the immunization we used the influenza virus accumulated in developing chicken embryos, but for the ELISA the plates were covered with the virus obtained on MDCK cell culture (information is added in the “materials and methods” section (line 122)). Thus, the results reflect the virus-specific activity of serum IgG. Serum from non-immunized mice was used as a negative control.

Answer to Q6. Mice, used for humoral response analysis (section 3.1, Fig.1A) were later infected with A/Aichi/2/68 for viral load measurement on day 2 and 4 after the challenge (section 3.2, Fig.2C). A separate immunization experiment with the same conditions was performed to determine T-cellular response in mouse lungs and spleen (section 3.1, Fig.1B, C).

Answer to Q7a. The X-axis represents the days of the experiment. Changes are introduced in Figure 2C.

Answer to Q7b. The data represented in Figure 2A are obtained from 6 animals, as line 199 states.

Answer to Q8. The growing evidence from recent researches supports the view that the long-lasting reactions of the innate immune system should be considered during the vaccine design and clinical trials as well as the adaptive immune response features.

In this work we used inbred C57/Black-6 mice with known MHC-haplotype to measure T-cellular immune response to immunodominant influenza CD8 epitope. The suggestions on the animal strain selection would be taken into account for the design of future investigations.

Reviewer 3 Report

The NS1 protein of influenza virus is an antagonist of the innate immune system that can stimulate an excessive immune response that is harmful to the host.  Therefore, it is of interest to develop vaccine strategies that attenuate the inflammatory immune response.  In this manuscript, Vasilyev  et al. tested the effectiveness of intranasal immunization with an influenza A virus encoding a NS1 expressing only the N-terminal half of the NS1 protein (124aa).  It was demonstrated that mice immunized with the NS-1 shortened virus (A/PR8/NS124) were better protected from challenge with the heterologous virus.  Moreover, immunization of with A/PR8/NS124 induced an enhanced influenza-specific CD8+ T cell responses in the lungs and was associated with decreased proinflammatory cytokine concentrations and decreased leukocyte infiltration.  This study suggests  that the NS1 shortened virus conferred better protection from lethality after challenge with heterologous virus, while reducing the severity of the innate immune response.   This is a well-executed study that shows the advantages of the NS1 truncated virus as a vaccine that confers protection from virus infection by inducing both humoral and T cell-mediated immunity while also preventing excessive inflammation.

Minor concern:

  1. Figure 6 is very confusing to read through. The figure should be broken into “A” Box plot “B”  pie chart and C.  Bar graph.  The bar graph is  broken into groups of three with orange blue, green, pink, darkblue, light green and red bars underneath.  It is unclear what the bar colors correspond to or if it just a way of designating the groups.  This should be explained more clearly.  If it is just to break up the groups of three, a space or line between the groups might be easier for the eye to navigate.

Author Response

Thank you for your suggestions. Additional clarifications are introduced in captions of figures 1 and 4 (line 156-160; line 300-305):

Histograms represent the percentage of different populations of cytokine-producing T-cells in the total effector memory (EM) CD8+CD44+CD62L- T-lymphocytes population. Pie charts represent the percentage of cells producing any combination of IFNγ, IL2, or TNFα cytokines in the total cytokine-producing CD8+ EM T-cell subset. Color bars under histogram mark different cytokine-producing populations and correspond to pie chart color scheme.